# Expression of Prooncogenic Nuclear Receptor 4A (NR4A)-Regulated Genes β1-Integrin and G9a Inhibited by Dual NR4A1/2 Ligands

**DOI:** 10.3390/ijms26083909

**Published:** 2025-04-21

**Authors:** Lei Zhang, Victoria Gatlin, Shreyan Gupta, Michael L. Salinas, Selim Romero, James J. Cai, Robert S. Chapkin, Stephen Safe

**Affiliations:** 1Department of Veterinary Physiology and Pharmacology, Texas A&M University, College Station, TX 77843, USA; lzhang@cvm.tamu.edu; 2Department of Veterinary Integrative Biosciences, School of Veterinary Medicine and Biomedical Sciences, Texas A&M University, College Station, TX 77843, USA; leeva808@exchange.tamu.edu (V.G.); xenon8778@exchange.tamu.edu (S.G.); ssromerogon@tamu.edu (S.R.); jcai@tamu.edu (J.J.C.); 3CPRIT Single Cell Data Science Core, Texas A&M University, College Station, TX 77843, USA; mlsalinas4@tamu.edu (M.L.S.); robert.chapkin@ag.tamu.edu (R.S.C.); 4Department of Nutrition, Texas A&M University, College Station, TX 77843, USA; 5Department of Electrical and Computer Engineering, Texas A&M University, College Station, TX 77843, USA

**Keywords:** NR4A1, NR4A2, pro-oncogenic, targetable

## Abstract

Bis-indole-derived compounds including 1,1-bis(3′-indolyl)-1-(3,5-disubstitutedphenyl)methane (DIM-3,5) analogs bind both orphan nuclear receptors 4A1 (NR4A1) and NR4A2, and DIM-3,5 compounds act as dual receptor inverse agonists and inhibit both NR4A1- and NR4A2-regulated responses. Chromatin immunoprecipitation assays show that β1-integrin and the methyltransferase gene G9a are regulated by both NR4A1 and NR4A2 acting as cofactors for Sp1- and Sp4-dependent gene expression. DIM-3,5 treatment results in the loss of one or more of these nuclear factors from the β1-integrin and G9a promoters. Single-cell and RNAseq analyses show that both receptors regulate common (<10%) and unique genes in SW480 colon cancer cells; however, functional enrichment analysis of the differentially expressed genes converges to several common pathways and gene ontology terms.

## 1. Introduction

Colorectal cancer (CRC) is a highly complex disease with multiple risk factors including both genetic/heritable and environmental components that contribute to disease incidence [1,2,3,4,5]. Despite increased participation in screening techniques and the development of new treatment regimens, there are over 1.8 million new cases and 900,000 deaths per year from CRC worldwide [6,7]. Cases of CRC are highest in the more developed countries such as the United States, Canada and European countries and tend to increase in less affluent countries during periods of their economic growth and development [6,7]. Individuals with a family history of colon cancer or inherited genetic mutations have a high risk for developing CRC; however, it is estimated that 60–65% of all cases are “sporadic” with no inherited or genetic risk factors in their background [8,9,10]. Hereditary cancer syndromes linked to colon cancer include hereditary nonpolyposis colorectal cancer (HNPCC, Lynch Syndrome) and familial adenomatous polyposis (FAP), and these are caused by mutations in high-penetrance genes. For example, patients with FAP have APC mutations and continually develop multiple adenomas in the distal colon that must be removed or their risk of colon cancer will be 100% by 40 years of age [11,12,13]. Some of the risk factors for 60–65% of sporadic CRCs are shared by many other cancers, and this includes age but also other preventable factors such as obesity, sedentary lifestyle, smoking, low-fiber/high-fat diets, alcohol consumption and high intakes of red meat [14,15,16,17,18,19,20]. Another sub-class of CRC patients are young adults, and the incidence of disease among this group has been increasing, particularly in high income developed countries [21,22]. Although many of the same risk factors apply for young adults with sporadic CRC, the reasons for the early onset of this disease are not well defined. The treatment of CRC includes surgeries, radiation and chemotherapy using drug combinations, and the selection of specific therapeutic options is dependent on multiple factors including the stage of the tumor. In addition, the development of precision medicine approaches targeting kinases (EGFR, HER2) angiogenesis (VEGF/VEGFR) and immunotherapeutics (e.g., antibodies) are being pursued to enhance therapeutic effectiveness.

Studies in our laboratory have focused on developing agents that target the orphan nuclear receptor 4A1 (NR4A1, Nur77) in solid tumors including colon cancer cells [23,24]. Results of NR4A1 knockdown studies show that this receptor is pro-oncogenic in solid tumors and regulates cancer cell proliferation, survival, migration and invasion and plays a role in T-cell exhaustion [25,26]. NR4A1 is overexpressed in colon tumors and is a negative prognostic factor for patient survival [27]. Studies in this laboratory have identified a series of 1,1-bis(3′-indolyl)-1-(substitutedphenyl)methane (CDIM) analogs that bind NR4A1 and act as inverse agonists to inhibit NR4A1-regulated pro-oncogenic pathways and genes [23]. Limited data suggest that, like NR4A1, NR4A2 is also a pro-oncogenic factor in solid tumors, and in colon cancer, NR4A2 overexpression predicts an unfavorable prognosis and chemoresistance [28]. Recent studies have shown that a series of 1,1-bis(3′-indolyl)-1-(3,5-disubstituedphenyl)methane (DIM-3,5) analogs were identified as NR4A1 ligands that are potent inhibitors of breast cancer growth in an orthotopic athymic nude mouse model with IC_50_ values for tumor growth inhibition of <1 mg/kg/day [29]. This high anticancer activity suggested that other factors in addition to NR4A1 might be involved, and we recently reported that DIM-3,5 analogs were NR4A1/NR4A2 ligands that bound both receptors and inhibited cancer cell growth [30]. In this study, the activities of 3 DIM-3,5 analogs and their effect on NR4A1 and NR4A2 as dual NR4A1/NR4A2 ligands were investigated, and we show that like NR4A1, NR4A2 exhibits pro-oncogenic activity in colon cancer cells, and the compounds act as dual-receptor inverse agonists that inhibit NR4A1-/NR4A2-regulated responses.

## 2. Results

In this study, we used three DIM-3,5 compounds, which include the 3,5-dichlorophenyl (DIM-3,5), 3,5-dibromophenyl (DIM-3,5-Br_2_) and 3-chloro-5-trifluoromethylphenyl (DIM-3-CI-5-CF_3_) analogs, exhibiting similar KD values for binding NR4A2 (5.5–12.0 µM) and lower K_D_ values for binding NR4A1 (3.1–7.7 µM) [30]. Results illustrated in Figure 1A show that the three DIM-3,5 analogs decrease the growth of RKO cells, and significant growth inhibition is observed at concentrations ranging from 5 to 10 µmol/L. Similar results were observed in SW480 (Figure 1B) and HCT116 (Figure 1C) colon cancer cells, and DIM-3,5-Br_2_ was the least active compound in RKO and HCT116 cells. The effects of these compounds on NR4A1 and NR4A2 expression and on previously identified NR4A1-responsive genes were also determined in RKO and SW480 cells. The DIM-3,5 analogs did not affect NR4A2 expression in either cell line (Figure 1D,E), whereas 7.5 or 15 µmol/L of DIM-3,5 compounds decreased levels of NR4A1 protein. Both Sp1 and Sp4 form complexes with NR4A1 (i.e., NR4A1/Sp) to regulate expression of some genes [23], and the DIM-3,5 dual NR4A1/2 ligands decreased levels of Sp1 and Sp4 proteins. DIM-3,5 compounds also decreased expression of NR4A1-responsive gene products G9a, β1-integrin and bcl-2 [23,24,25] in RKO and SW480 cells (Figure 1F,G), and Parp cleavage was also observed. The quantitation of Western blot results is summarized in Figure 1H,I. The effects of the knockdown of NR4A1, NR4A2 or their combination was also investigated, and Figure 2A shows that oligonucleotides targeting NR4A1 (siNR4A1) and NR4A2 (siNR4A2) decreased RKO cell viability, with comparable results observed in SW480 (Figure 2B) and HCT116 (Figure 2C) cells. Figure 2D illustrates the differential effectiveness of the oligonucleotides in downregulating NR4A1 and NR4A2, and SW480 and RKO cells were selected for cell proliferation (Figure 2A–C) and Western blot (Figure 2E) experiments. RNA interference was also used to examine the effects of siNR4A1 and siNR4A2 on Sp1, Sp4 and several prototypical NR4A1-responsive genes (Figure 2E). The results show that both siNR4A1, siNR4A2 and their combination decrease expression of these gene products, suggesting that G9a, bcl2 and β1-integrin are co-regulated by both NR4A1 and NR4A2. In addition, we also observed that the knockdown of NR4A1 and NR4A2 decreased expression of Sp1 and Sp4 (Figure 2A). Previous studies have shown that NR4A regulates pro-reductant genes and the knockdown of NR4A1 or treatment with CDIMs induces ROS [23], and ROS activate pathways that downregulate Sp transcription factors. This has also been observed for other ROS-inducing compounds which decrease expression of Sp and Sp-regulated genes [31,32].

The effects of DIM-3,5-Br_2_, DIM-3-CI-5-CF_3_ and DIM-3,5-CI_2_ on NR4A1- and NR4A2-dependent transactivation were determined in RKO (Figure 3A), SW480 (Figure 3B) and HCT116 (Figure 3C) colon cancer cell lines transfected with GAL4-NR4A1 or GAL4-NR4A2 chimeras and UAS-Luc, which contains five yeast GAL4 response elements. The three CDIM compounds decreased transactivation in RKO cells expressing GAL4-NR4A1, and this is consistent with the inverse NR4A1 agonist activities observed for these compounds in inhibiting expression of various gene products. In contrast, the three CDIM compounds induced luciferase activity in RKO cells expressing GAL4-NR4A2. Although this enhanced transactivation did not parallel the observed growth inhibition by DIM-3,5 compounds or NR4A2-dependent downregulation of gene products, the results are comparable with previous studies using GAL4-NR4A2 and other NR4A2 ligands. The GAL4-NR4A2-dependent transactivation is cell context-dependent and is induced in colon and pancreatic cancer cells [30,33] and decreased in U87MG glioblastoma cells [30], whereas CDIM compounds that bind NR4A2 decrease cell viability in colon, pancreatic cancer and glioblastoma cells [23,24,25,30]. The pattern of inhibited/induced transactivation observed for the effects of these dual NR4A1/2 ligands in RKO cells was also observed in SW480 (Figure 3B) and HCT116 (Figure 3C) colon cancer cell lines and confirms that the CDIM compounds target both NR4A1 and NR4A2 in transactivation assays using GAL4-receptor chimeras but with inverse induction responses in colon cancer cells.

A comparison of the effects of NR4A1 and NR4A2 knockdown and the DIM-3,5 dual NR4A1/NR4A2 ligands on colon cancer cell migration and apoptosis (TUNEL assay) was also investigated in SW480 cells. Compared to the untreated or scrambled non-specific oligonucleotide, siNR4A1 and siNR4A2 alone or in combination, with 7.5 or 15 µmol/L DIM-3,5-CI_2_, inhibited SW480 cell migration (Figure 4A). A comparable approach was also taken in SW480 cells using the TUNEL assay for apoptosis and both the knockdown of NR4A1 and NR4A2 and their combination and DIM-3,5-CI_2_ induced TUNEL staining in SW480 cells (Figure 4B). Previous studies showed that both β1-integrin and G9a were NR4A1-responsive genes [23] and results in Figure 1 confirm that both genes are regulated by NR4A1 and NR4A2. It was previously reported that NR4A1 acted as a cofactor and NR4A1/Sp1 and NR4A1/Sp4 regulated β1-integrin and G9a, respectively, by binding their corresponding GC-rich promoter sequences. This mechanism where NR4A1 acts as a ligand-dependent cofactor to induce or repress gene expression has been previously observed for many other nuclear receptors [33]. Oligonucleotides targeting Sp1 (siSp1) and Sp4 (siSp4) were transfected into RKO and SW480 cells and both Sp1 and Sp4 decreased expression of G9a and β1-integrin in RKO and SW480 cells (Figure 5A). This suggested that Sp1 and Sp4 are also involved in the regulation of G9a, and β1-integrin expression and RKO cells were used as a model for further investigating the mechanisms of β1-integrin and G9a regulation. Figure 5B,C summarize the results of ChIP assays showing relative levels of NR4A1, NR4A2, Sp1 and Sp4 associated with an NR4A1-responsive GC-rich region of the β1-integrin promoter after treatment with DIM-3,5-CI_2_. All four proteins are detected and there are some treatment-related changes such as decreased levels of NR4A2 and Sp1. The overall model (Figure 5C) shows interactions of both NR4A1 and NR4A2 with Sp1/4 on the β1-integrin promoter, and it is also possible that NR4A1 and NR4A2 bind Sp as monomers or homodimers, as previously suggested in studies on NR4A1 interactions with the β1-integrin promoter [34]. However, since it has previously been reported that NR4A1 and NR4A2 can also form heterodimers [35], it is possible that the NR4A1-NR4A2 heterodimer binds Sp1 or Sp4. Result obtained in a ChIP assay on the G9a promoter were similar to those obtained for β1-integrin; however, DIM-3,5-CI_2_ treatment decreased interactions of NR4A1, NR4A2, Sp1 and Sp4 with the GC-rich region of the G9a promoter (Figure 5D). These results are consistent with the coregulation of G9a and β1-integrin by NR4A1 and NR4A2, and this complements the results of functional assays where both NR4A1 and NR4A2 exhibit pro-oncogenic activity that is inhibited by dual DIM-3,5 inverse NR4A1/NR4A2 agonists.

In complementary experiments, the genomic effects of NR4A1 and NR4A2 were also investigated in SW480 cells using RNA interference and subsequent scRNA-seq analysis (Section 4). Figure 6A,B summarizes the effects of NR4A1/NR4A2 knockdown and DIM-3,5-CI_2_ on NR4A1 and NR4A2 mRNA levels in SW480 cells. Figure 6C illustrates a UMAP visualization of the single cells according to their group/treatment identity. The results in Figure 6D,E show that DIM-3,5-CI_2_ decreased the expression of NR4A1, c-parp, β1-integrin, G9a, Sp1 and Sp4 but increased NR4A2 mRNA levels in SW480 cells. DE analysis was performed using the default testing implemented in the R Seurat package (Section 4). DE genes were identified by comparing the DMSO control group versus each of the other groups, including transfection with siRNAs individually targeting NR4A1 or NR4A2, versus combined NR4A1 and NR4A2 siRNAs or treatment with DIM-3,5-CI_2_. Overlap of identified up- and downregulated DE genes between different comparison pairs is shown in Venn diagrams (Figure 7A,B). The numbers of overlapping DE genes between different pairwise group comparisons were small, ranging from 14 (downregulated DE genes shared between control versus DIM-3,5-CI_2_ and control versus siNR4A1) to 77 (upregulated DE genes shared between control versus DIM-3,5-CI_2_ and control versus siNR4A2) out of the top 500 genes in each group. The limited overlap suggests relatively independent effects at the individual gene level in the siRNA knockdown and DIM-3,5-CI_2_ treatments in SW480 cells. However, at the pathway level, the results of functional enrichment analysis with these DE genes were converged to several specific pathways and gene ontology terms, including, for example, *Apoptotic Signaling Pathway*, *Epithelial Cell* or *Fibroblast Proliferation* and *MAP Kinase Activity* (Figure 7C). Thus, the pathway-level analysis suggests that the siRNA system and DIM-3,5-CI_2_ treatment produced a similar functional impact on gene expression programs in SW480 cells.

## 3. Discussion

NR4A subfamily members NR4A1, NR4A2 and NR4A3 were first identified as immediate early response genes, and subsequent studies show that they regulate important functions/genes in maintaining cellular homeostasis and pathophysiology [36,37]. The dual NR4A1 and NR4A3 knockout mice rapidly develop acute myeloid leukemia [38], and in blood-derived cancers, NR4A1 and NR4A3 exhibit tumor suppressor-like activity. In contrast, in most solid tumors, both NR4A1 and NR4A2 exhibit pro-oncogenic activity, whereas, in limited studies, NR4A3 exhibits either minimal activity [23] or acts as a tumor suppressor [39]. In several cancer- and non-cancer-related diseases, NR4A1 and NR4A2 are over-expressed and are targetable with receptor agonists or inverse agonists to ameliorate disease, and there are increasing studies on development of new NR4A ligands [23,30].

In this laboratory, a series of bis-indole-derived compounds were developed and 1,1-bis(3′-indolyl)-1-(4-hydroxy phenyl)methane (DIM-4-OH) and 1,1-bis(3′-indolyl)-1-(4-chlorophenyl)methane (DIM-4-CI) were used as prototypical first-generation NR4A1 and NR4A2 ligands, respectively [38]. DIM-4-OH interacted directly with the ligand binding domain (LBD) of NR4A1 [24], and modeling studies showed that DIM-4-CI preferentially interacted with a cofactor site of NR4A2 [23,38]. Subsequent structure–activity studies identified a series of DIM-3,5 analogs that were significantly more potent as inhibitors of tumor growth in athymic nude mice bearing MDA-MB-231 breast cancer cells (orthotopic) [29]. IC_50_ values of <1 mg/kg/day were observed in this study, which included DIM-3-CI-5-CF_3_- and DIM-3,5-CI_2_ (2.5 mg/kg/day)-inhibited colon tumor growth and reversed T-cell exhaustion in a syngeneic mouse model using MC38 cells as a xenograft [25]. The high in vivo potencies of the DIM-3,5 analogs coupled with the >60% similarity of the LBDs of NR4A1 and NR4A2 suggested that the DIM-3,5 compounds may bind and inactivate both receptors, and this was confirmed in a recent publication [30].

In this study, the functional and genomic properties of NR4A1 and NR4A2 are contrasted through the knockdown of each receptor by RNA interference, and the results are compared to that of the DIM-3,5 analogs, which are dual NR4A1/NR4A2 ligands. The results of NR4A2 knockdown in colon cancer cells confirmed the pro-oncogenic activity of this gene and was similar to that previously observed for NR4A2 other cancer cell lines [23] and consistent with the unfavorable prognosis for colon cancer patients that overexpress NR4A2. Overall, the results show that NR4A1 and NR4A2 exhibit similar pro-oncogenic activity and that the DIM-3,5 compounds are dual NR4A1/NR4A2 inverse agonists that induce responses similar to that observed for receptor silencing. Previous studies showed that β1-integrin and the histone methyltransferase G9a (EHMT2) gene were regulated by NR4A1 [23,34] and our results (Figure 2 and Figure 5) demonstrate that both β1-integrin and G9a are coregulated by both receptors that function as cofactors to enhance Sp1/Sp4-mediated transcription. Both NR4A1 and NR4A2 are associated with GC-rich regions of the β1-integrin and G9a gene promoters and it is possible that their cofactor activity may be due to these receptors acting as monomers, homodimers or even heterodimers, since it was previously reported that both NR4A1 and NR4A2 interact [35]. Moreover, dual NR4A1/NR4A2 regulation of TWIST1 in glioblastoma cells has also recently been reported using a similar approach [40].

We also used scRNA-seq to analyze the differentially expressed genes in SW480 cells after treatment with DIM-3,5-CI_2_ (24 h) or the knockdown of NR4A1, NR4A2 and NR4A1 plus NR4A2 (combined). mRNA levels observed in this analysis (Figure 6) were consistent with effects of DIM-3,5-CI_2_ treatment and NR4A1/2 knockdown on several gene products (Figure 1, Figure 2 and Figure 5). The Venn diagram analysis of the results showed that in the various treatment groups, several DE genes were commonly induced or repressed; however, it was also apparent that each treatment group induced or repressed many genes independently. These analyses were from a single timepoint (24 h treatment) and knockdown (72 h) experiments, and the overlap of DE genes may be time-dependent and requires further investigation. Nevertheless, pathway-level comparisons using the results from functional enrichment analysis of DE genes showed that all treatment groups (siRNAs and DIM-3,5-CI_2_) converged on common pathways (Figure 7C). Current studies are applying these same approaches using DIM-3,5-CI_2_ and its effects on tissue specific knockdown of NR4A1 and NR4A2 and comparing results of DE genes in vivo and in cell culture.

## 4. Materials and Methods

### 4.1. Cell Culture, Reagents, Antibodies

Three colon cancer cell lines RKO (RRID:CVCL_0504), SW480 (RRID:CVCL_0546) and HCT116 (RRID:CVCL_0291) were purchased from American Type Culture Collection (Manassas, VA, USA). Cells were cultured in DMEM medium with 10% FBS at 37 °C in the presence of 5% CO_2_. The primary antibodies used for Western blot were as follows:

Cleaved PARP (5625S), Bcl-2 (15071S), G9a (68851) and β1-integrin (9699S) were from Cell Signaling Technology (Danvers, MA, USA); NR4A2 (sc-376984X), Sp1 (sc-17824X) and Sp4 (sc-390124X) were from Santa Cruz Biotechnology (Dallas, TX, USA); NR4A1 (ab283264) was purchased from Abcam (Waltham, MA, USA); β-actin (A1978) was from Sigma Aldrich Corporation (Milwaukee, WI, USA); and secondary antibodies for rabbit (7074) and mouse (7076) studies were purchased from Cell Signaling Technology (Danvers, MA, US). The following antibodies were used for ChIP assays: NR4A1 (sc-365113X), NR4A2 (sc-376984X), Sp1 (sc-17824X), Sp4 (sc-390124X) and IgG (sc-515946), purchased from Santa Cruz Biotechnology (Dallas, TX, USA).

### 4.2. Cell Proliferation Assay

Cell proliferation was investigated using the XTT Cell Viability Kit (Cell Signaling Technology) according to the manufacturer’s instructions. Cells (1.5 × 10^4^/well) were seeded in DMEM medium with 10% FBS on 96-well plates and allowed to attach for 24 h. The complete medium was then changed to DMEM containing 2.5% charcoal-stripped FBS, and either vehicle (dimethyl sulfoxide, DMSO), specific concentrations of compounds in DMSO, or targeted siRNAs were added. After 24 h of drug treatment or 72 h after transfection of siRNAs, 35 μL of XTT reaction [solution (sodium3′-[1-(phenyl-aminocarbonyl)-3,4-tetrazolium]-bis(4-methoxy-6-nitro) benzenesulfonic acid hydrate and N-methyl dibenzopyrazine methyl sulfate; mixed in proportion 50:1) was added to each well. The optical density was read at a 450 nm wavelength in a plate reader after 4 h of incubation. All determinations were replicated in at least three separate experiments.

### 4.3. Transfection and Luciferase Assay

Cells were seeded on 12-well plates at 5 × 10^4^ cells/well in DMEM medium supplemented with 2.5% charcoal-stripped FBS. After 24 h, various amounts of plasmid DNA [i.e., UASx5-Luc (400 ng), GAL4-NR4A1 (250 ng) and β-gal (250 ng)] were cotransfected into each well by GeneJuice Transfection reagent (Millipore Sigma, Darmstadt, Germany) according to the manufacturer’s protocol. After 6 h of transfection, cells were treated with plating media (as indicated above) containing either solvent (DMSO) or the indicated concentration of compound for 18 h. Cells were then lysed using a freeze–thaw protocol, and 30 μL of cell extract was used for determining luciferase (E1483, Promega, Madison, WI, USA) and β-gal (T1007, Invitrogen, Waltham, MA, USA) activities. LumiCount (Packard, Meriden, CT, USA) was used to quantify luciferase and β-gal activities. Luciferase values were normalized against corresponding β-gal activity as well as protein concentrations determined in a Bradford assay.

### 4.4. Transfection and Small Interfering RNAs

For RNA interference experiments, cells were seeded on a 6-well plate at 1.5 × 10^5^ cells/well then allowed to attach for 24 h. Cells were then transfected with siRNAs (100 nmol each/well in 6-well plates) using 6.5 μL/well RNA iMax transfection reagent. After 72 h, cell lysates were obtained. Small interfering RNAs (siRNAs) targeting NR4A1 (siNR4A1), NR4A2 (siNR4A2), Sp1 (siSp1) and Sp4 (siSp4) were purchased from Sigma-Aldrich (St. Louis, MO, USA) and Integrated DNA Technologies (IDT) (Coralville, IA, USA). Negative Control Ig L2 siRNA were purchased from Qiagen (Montreal, ON, Canada). The oligonucleotides used were as follows:

siNR4A1_1, Sigma, SASI_Hs02_00333289

siNR4A1_2, Sigma, SASI_Hs02_00333290

siNR4A2_1, IDT, 5′-/52MOErA/*/i2MOErA/*/i2MOErG/*/i2MOErA/*/i2MOErT/*G*A*G*T*T*T*A*C*C*C*T*/i2MOErC/*/i2MOErC/*/i2MOErA/*/i2MOErC/*/32MOErT/-3′

siNR4A2_2, Sigma, EHU008731

siSp1_1: Sigma, SASI_HS01-00070994

siSp1_2: Sigma, SASI_HS02_00333289

siSp4_1: Sigma, SASI_HS01-00114420

siSp4_2: Sigma, SASI_HS01-00114421

### 4.5. Scratch Migration Assay

Attached cells (2.0 × 10^5^) were treated with different concentrations of specific compounds or after transfection with siRNAs in DMEM medium supplemented with 2.5% charcoal-stripped FBS and grown to 90% confluency in 6-well plates; cells were then scratched with a 200 μL sterile pipette tip and washed with PBS to remove detached cells from the plates. Cells were kept in an incubator for 48 h and were then fixed with 4% formaldehyde solution and stained with crystal violet solution. The wound gap was observed under an AMG EVOS fluorescence microscope. At least 3 replicates were performed for each treatment group.

### 4.6. TUNEL Assay

A TUNEL assay was performed using a One-Step TUNEL In Situ Apoptosis Kit (Green, FITC) (E-CK-A320, Elabscience, Houston, TX, USA) according to the manufacturer’s protocol. Cells (1.0 × 10^5^) were seeded on 12-well plates covered with glass slides and allowed to attach for 24 h. After compound treatments for 24 h transfection with siRNAs for 72 h, cell slides were fixed in the fixative buffer (4% of polyformaldehyde in PBS) at 4 °C for 1.5 h then washed (3×) for 5 min each, and cell slides were immersed in the permeabilization buffer (0.2% of Triton-100 in PBS) at 37 °C for 10 min. After permeabilization was finished, cells were washed (3×) for 5 min and the labeling protocol was carried out as recommended by the manufacturer.

### 4.7. Western Blot Analysis

Cells (3.0 × 10^5^) were seeded on a 6-well plate and allowed to attach for 24 h. After treatment with DIM-3,5 compounds for 24 h or siRNA transfection for 72 h, whole-cell lysates were obtained by treating them with a high-salt lysis buffer RIPA (Thermo Scientific, Waltham, MA, USA) containing protease and phosphatase inhibitors (GenDEPOT, Baker, TX, USA). The total protein in the lysates was quantified by a Bradford assay, and 25 µg of protein from each lysate were loaded on SDS polyacrylamide gels, transferred to a PVDF membrane with 90 volts for 1 h and then blocked for an hour using 5% skimmed milk. Membranes were then incubated with primary antibody overnight at 4 °C, then washed with Tris-buffered saline and Polysorbate 20 (TBST) and incubated with HRP-linked secondary antibody for 1 h at room temperature. The membranes were further washed with TBST and treated with Immobilon Western Chemiluminescence HRP Substrates (WBKLS0500, Millipore Sigma, Burlington, MA, USA) to detect the protein bands using the Bio-Rad Chemidoc MP Imaging System (BioRad, Hercules, CA, USA).

### 4.8. ChIP Assay

The chromatin immunoprecipitation (ChIP) assay was performed using the ChIP-IT Express magnetic chromatin immunoprecipitation kit (Active Motif, Carlsbad, CA, USA) following the manufacturer’s protocol. All cells (2 × 10^7^) were treated with DMSO or indicated concentrations of DIM-3,5 compounds for 24 h. Cells were then fixed with 1% formaldehyde, and the cross-linking reaction was stopped by the addition of 0.125 M glycine. After washing twice with phosphate-buffered saline, cells were scraped and pelleted. Collected cells were hypotonically lysed, and nuclei were collected. Nuclei were then sonicated to the desired chromatin length (200 to 1500 bp). The sonicated chromatin was immunoprecipitated with antibodies and protein G-conjugated magnetic beads at 4 °C overnight. After the magnetic beads were extensively washed, protein–DNA cross-links were reversed and eluted. Reversed cross-link DNA was prepared by proteinase K digestion followed by Chromatin IP DNA purification (Active Motif). The purified DNA products were analyzed by quantitative real-time PCR using the amfiSure qGreen Q-PCR master mix (genDEPOT) according to the manufacturer’s protocol. The primers for detection of the G9a promoter region were F—5′-CAG ATG GGG ACA GAG ACG C-3′ and R—5′-CCC GGA GCA TTG CAC G-3′, and, for detection of the β1 integrin promoter region, they were F—5′-TCA CCA CCC TTC GTG ACA C-3′ and R—5′-GAG ATC CTG CAT CTC GGA AG-3′.

### 4.9. Single-Cell RNA Sequencing (scRNA-seq) Analysis

For all samples, cell suspensions were collected and processed for single-cell isolation and RNA library preparation. Approximately 10,000 cells per sample were captured and used for library construction using the 10× Genomics Chromium Fixed RNA (Flex) Human Transcriptome Kit (PN-1000475, Genomics Chromium, Shirley, NY, USA). Library preparation was performed as per the manufacturer’s protocol. Sequencing was performed on the NovaSeq X Plus (Illumina, San Diego, CA, USA). The raw data (i.e., reads in FASTQ files) were mapped using CellRanger 8.0.0 referencing to the GRCh38-2024-A probe set. The expression matrices were imported into MATLAB (R2024b) scGEAToolbox (v24.1) (RRID:SCR_001622) [41] and R Seurat package [42] for subsequent data analysis. Stringent quality control filtering was applied to each sample requiring a minimum of 1000 reads per cell, a maximum mitochondrial DNA ratio per cell of 15%, a minimum of 15 non-zero cells per gene and a minimum of 500 non-zero genes per cell. DecontX [43] was applied to remove ambient RNA contamination. To plot the expression profile for individual genes, average gene expression values were imported into Prism software as a column data table, and non-parametric one-way ANOVA was used to compare between groups. The differential expression (DE) analysis was conducted using a non-parametric Wilcoxon rank sum test. Comparisons were made between the control and each of the treatment groups: DIM-3,5-CI_2_, siNR4A1, siNR4A2 and siNR4A1 and siNR4A2 (combined). Functional enrichment tests were conducted using the web interface of Enrichr [44] for the top 500 differentially expressed genes (ranked according to the *p* values).

### 4.10. Statistical Analysis

Each assay was performed in triplicate and the results were presented as means with standard deviations. The statistical significance of differences between the treatment groups was determined by Dunnett’s multiple comparison test in ordinary one-way ANOVA. Analysis of Western blots was carried out using ImageJ (1.53K) software (RRID:SCR_003070). GraphPad Prism 8 (Version 8.4.3) software (RRID:SCR_002798) was used for analysis of variance and determining statistical significance. Data with a *p*-value of less than 0.05 were considered statistically significant and indicated in the figures.

## Figures and Tables

**Figure 1 ijms-26-03909-f001:**
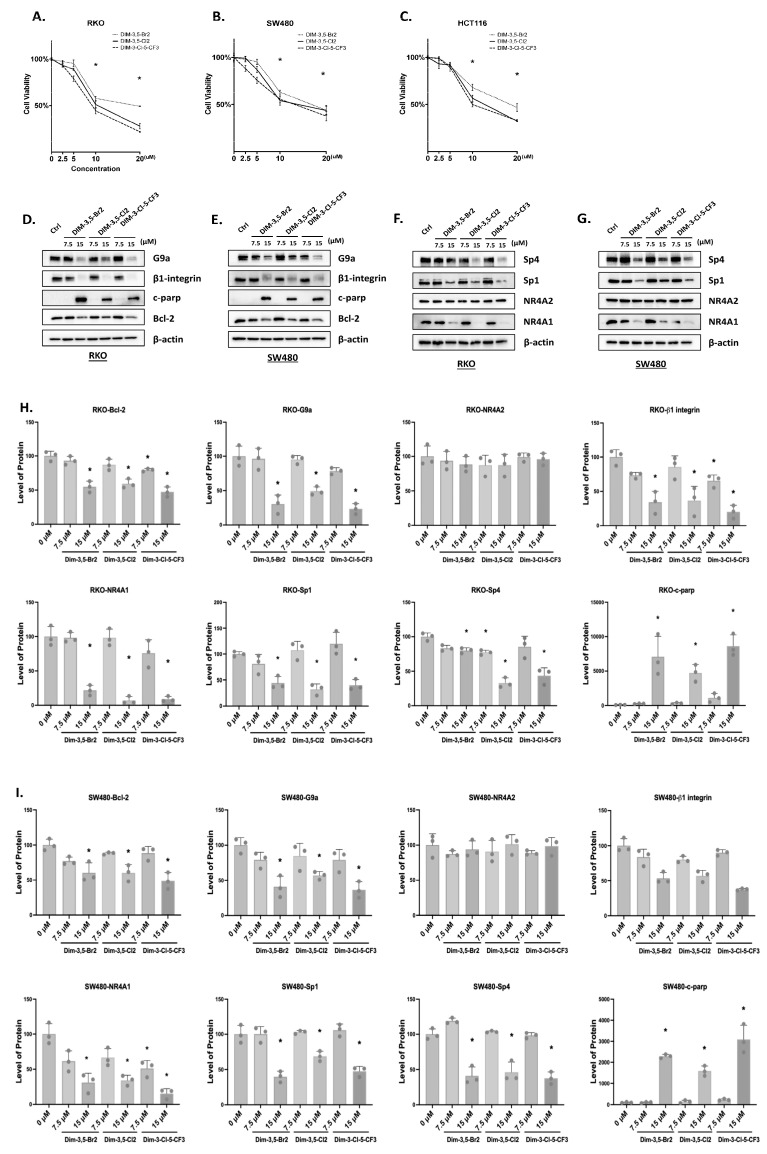
DIM-3,5 analogs inhibit cancer cell growth and modulate expression of NR4A1-regulated gene products. RKO (**A**), SW480 (**B**) and HCT116 (**C**) cells were treated with 2.5–20 µmol/L DIM-3,5-CI_2_, DIM-3-CI-5-CF_3_ and DIM-3,5-Br_2_ for 24 h and effects on cell proliferation were determined as outlined in Section 4. RKO and SW480 cells were treated with DIM-3,5 analogs for 24 h and whole-cell lysates were analyzed by Western blots (in triplicate), G9a, β1-integrin and apoptosis (**D**,**E**) and Sp/NR4A gene products (**F**,**G**). Results are expressed as means ± SE for at least 3 determinations for each treatment group and significant (*p* < 0.05) inhibition/induction compared to DMSO (control) is indicated (*) (**A**–**C**); quantitative results for (**D**–**G**) are summarized in (**H**,**I**) within the figure.

**Figure 2 ijms-26-03909-f002:**
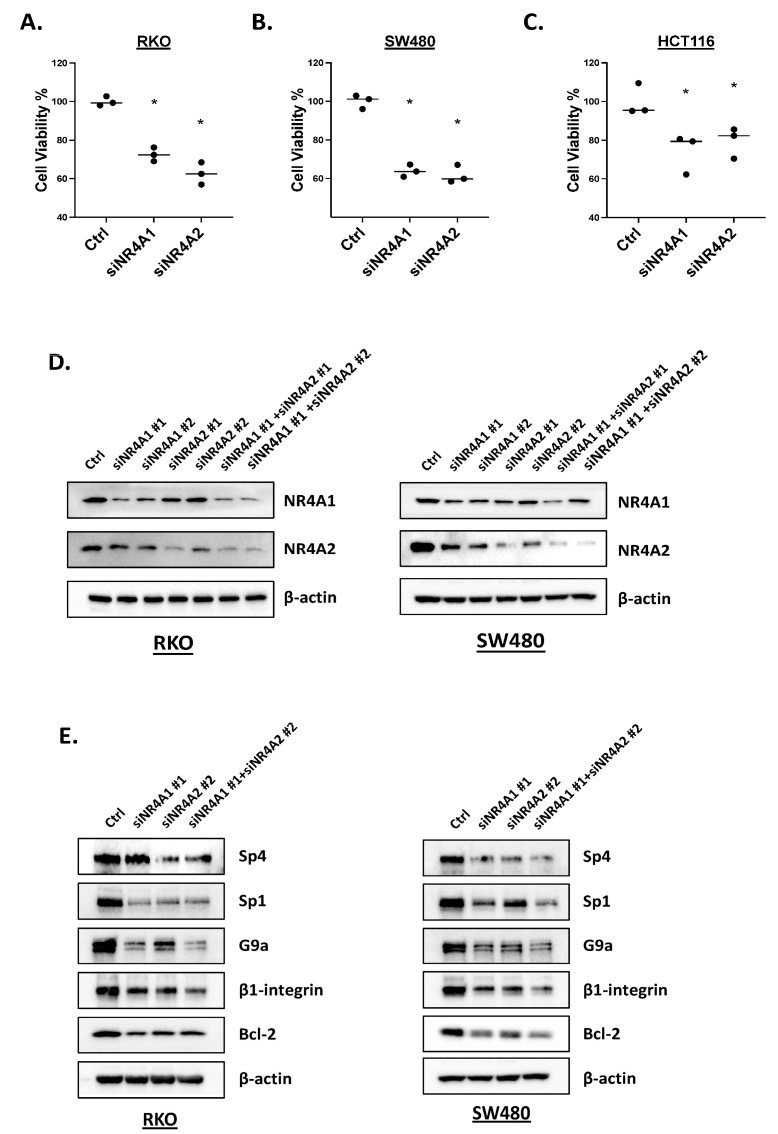
NR4A1 and NR4A2 knockdown on colon cancer cell growth and expression of selected genes. RKO, SW480 and HCT116 cells were transfected with oligonucleotides targeting NR4A1 (siNR4A1) and NR4A2 (siNR4A2) (**A**–**C**) and cell viability was determined. Multiple oligonucleotides targeting NR4A1 and NR4A2 and their combination were transfected into RKO or SW480 cells and whole-cell lysates were analyzed by Western blots knockdown to determine efficiency (**D**) and effects on selected gene products (**E**) as outlined in Section 4. Significant (*p* < 0.05) inhibition is indicated (*) in (**A**–**C**).

**Figure 3 ijms-26-03909-f003:**
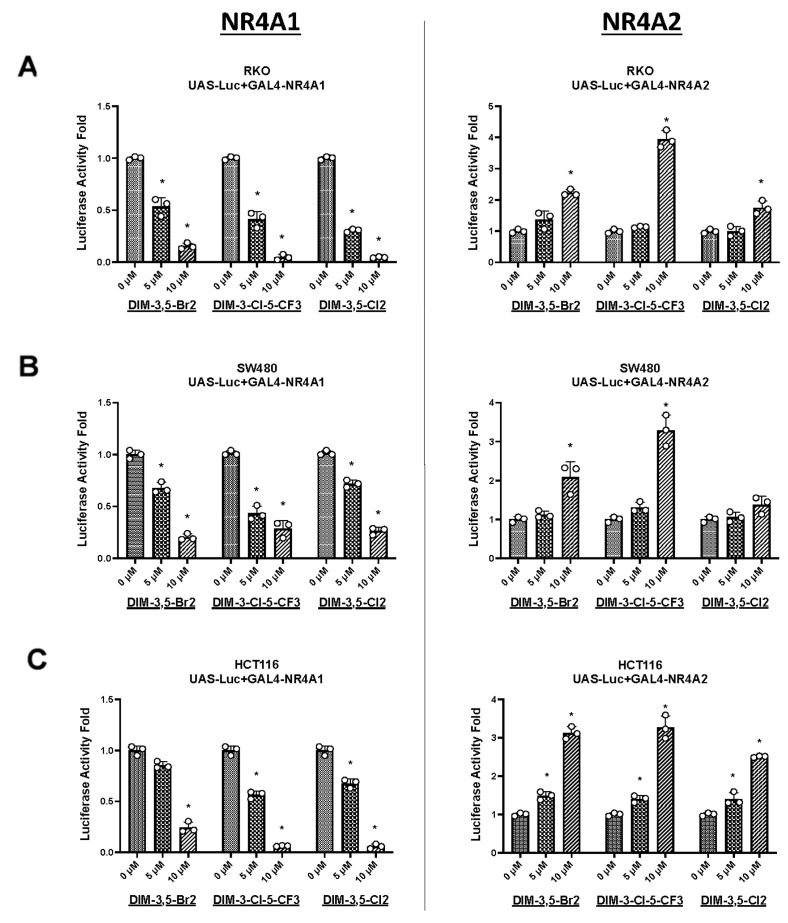
Effects of DIM-3,5 analogs on receptor-dependent transactivation. RKO (**A**), SW480 (**B**) and HCT116 (**C**) cells were transfected with GAL4-NR4A1 or GAL4-NR4A2 and a UAS-luc reporter gene and treated with DIM-3,5 analogs and luciferase activity was determined as outlined in Section 4. Results are expressed as means ± SE for at least 3 replicate determinations per treatment group, and significant (*p* < 0.05) induction/inhibition is indicated (*).

**Figure 4 ijms-26-03909-f004:**
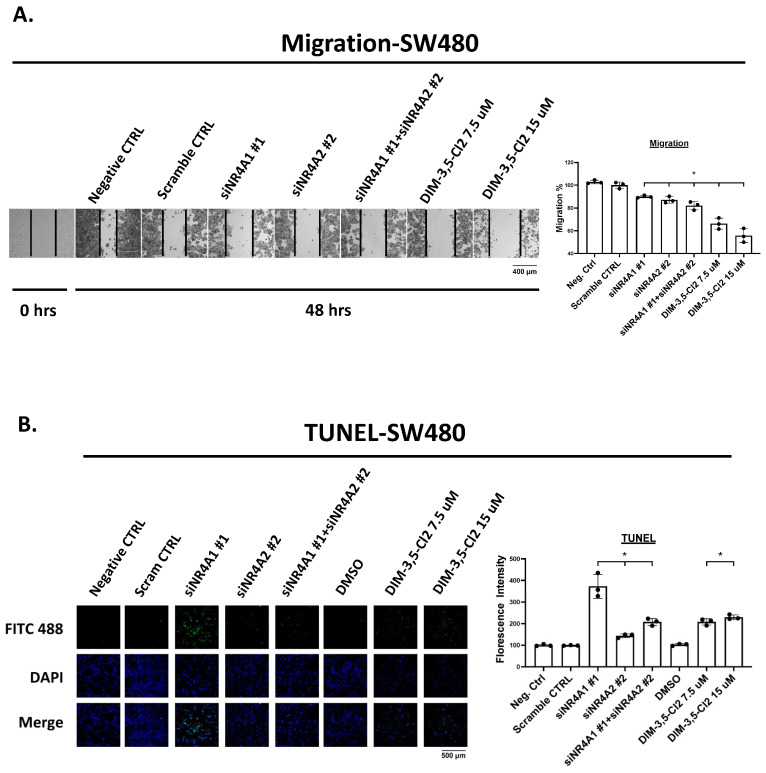
NR4A1 and NR4A2 knockdown and DIM-3,5 analogs inhibit invasion and induce TUNEL staining. SW480 cells were transfected with siNR4A1 and siNR4A2 or treated with DIM-3,5 analogs, and migration (**A**) and TUNEL (**B**) staining were determined as outlined in the Section 4 Results are expressed as means ± SE for at least 3 determinations per treatment group, and significant (*p* < 0.05) differences from the control group are indicated (*).

**Figure 5 ijms-26-03909-f005:**
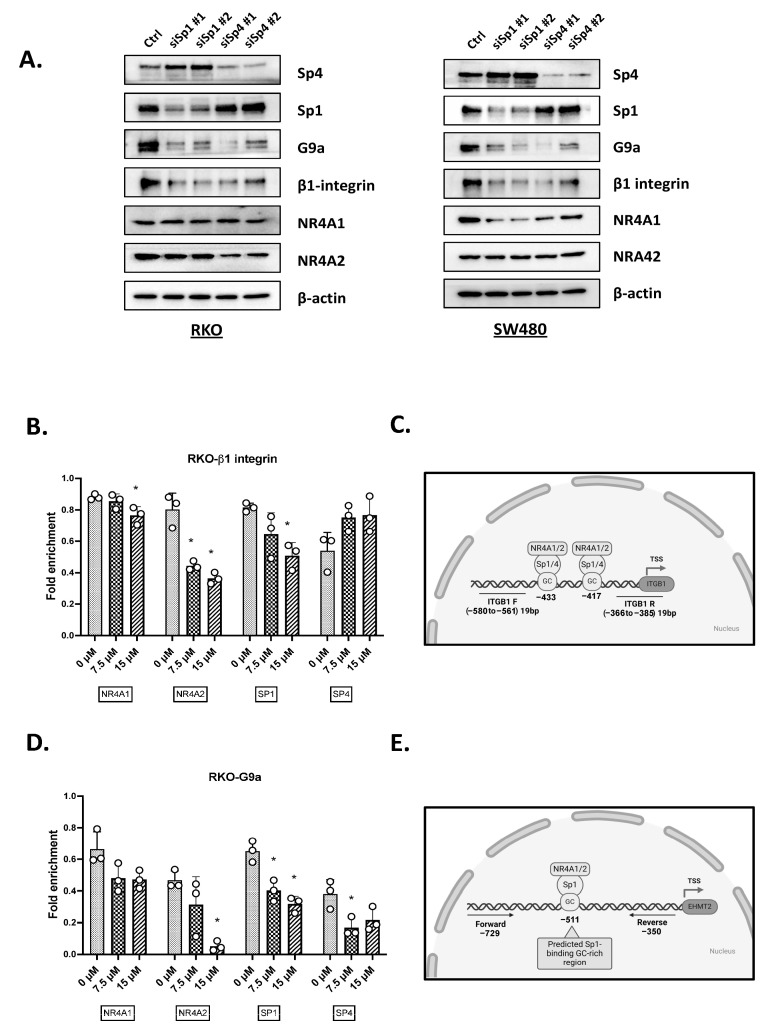
Effects of Sp1/Sp4 downregulation and the ChIP assay on the G9a and β1-integrin promoters in (**A**). (**A**) RKO and SW480 cells were transfected with oligonucleotides targeting Sp1 (siSp1) or Sp4 (siSp4) and whole-cell lysates were analyzed by Western blots as outlined in Section 4. RKO cells were treated with DIM-3,5-CI_2_ for 24 h and a ChIP assay was determined using primers for the β1-integrin (**B**,**C**) and G9a (**D**,**E**) promoters and interactions of NR4A1, NR4A2, Sp1 and Sp4 with the gene promoters was determined as outlined in the Section 4. Results (**B**,**D**) are means ± SE from 3 replicate experiments, and significantly (*p* < 0.05) decreased mRNA levels are indicated (*).

**Figure 6 ijms-26-03909-f006:**
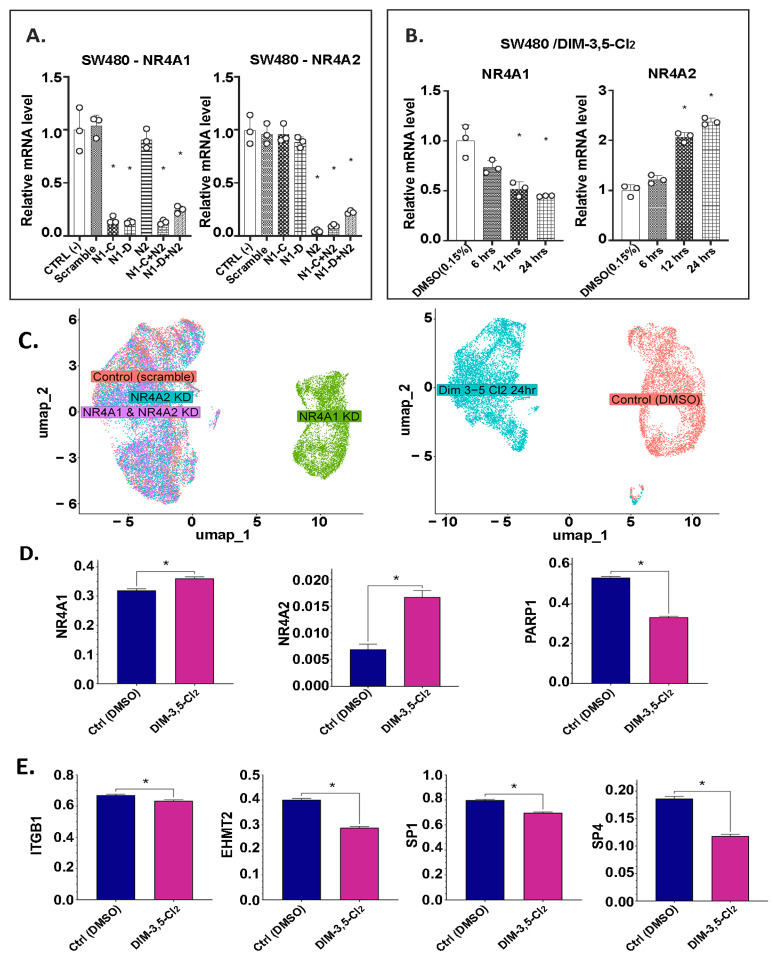
Results of scRNAseq analysis. Effects of knockdown (**A**) and DIM-3,5-CI_2_ treatment (**B**) on various mRNA levels by RT-PCR. (**C**) UMAP plots of the transcriptional landscape across individual cells. (**D**,**E**) Comparison of SW480 single-cell expression profiles, e.g., NR4A1, NR4A2, c-parp (PARP1), β1-integrin (ITGB1), G9a (EHMT2), Sp1 and Sp4 in control, Ctrl (DMSO) versus the DIM-3,5-CI_2_ treatment group. The small decrease in β1-integrin mRNA levels may be due, in part, to the single 24 h timepoint, since the decrease in gene expression varies over time. Significant (*p* < 0.05) effects are indicated (*) from triplicate determinations.

**Figure 7 ijms-26-03909-f007:**
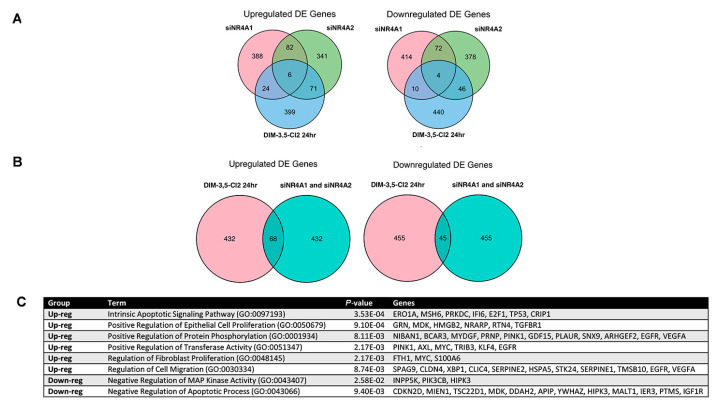
DE genes and pathways and their overlap. The overlap of DE genes identified when comparing the SW480 control versus transfected siRNAs individually targeting NR4A1 or NR4A2, versus treatment with DIM-3,5-CI_2_ (**A**), and combined NR4A1 and NR4A2 siRNAs (**B**). The top 500 up- or downregulated DE genes from each pair comparison are included in the Venn diagrams. (**C**) Consensus gene ontology terms and pathways enriched in DE genes. DE analysis was conducted between control, transfected or treated groups, i.e., control versus DIM-3,5-CI_2_, control versus siNR4A1, control versus siNR4A2 and control versus siNR4A1 and siNR4A2 (combined). DE genes identified between each of these pairs were subjected to functional enrichment analysis. Consensus gene ontology terms and pathways are those commonly identified.

## Data Availability

The data presented in this study are available on request from the corresponding author.

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
