# Peer review of "Expression of Prooncogenic Nuclear Receptor 4A (NR4A)-Regulated Genes β1-Integrin and G9a Inhibited by Dual NR4A1/2 Ligands"

_ijms, 2025, doi:10.3390/ijms26083909_

Round 1

Reviewer 1 Report

Comments and Suggestions for Authors

                The authors have performed an extensive body of work in investigating how DIM-3,5 compounds act as dual receptor inverse agonists and inhibit actions of the orphan nuclear receptors NR4A1 and NR4A2 in cell lines of colorectal carcinoma (CRC). DIM-3,5 compounds acted similarly and specifically in several CRC cell lines to reduce CRC cell viability and lower the protein levels of NR4A1 and the known targets beta1-integrin, Bcl2, and G9a. GAL4-luciferase and ChIP assays confirmed that these effects involved disruption of binding to target genes of the transcriptional activators Sp1 and Sp4. The findings indicate that NR4A1 and NR4A2 act as oncogenes in CRC, which may be inhibited by DIM-3,5 compounds or further-developed derivatives.

                The manuscript would be improved by attention to several points:

1) The authors should provide an explanation for why the protein levels of Sp1 and Sp4 are reduced by DIM-3,5 compounds or siRNA-mediated knockdown of NR4A1 and NR4A2. Although there are multiple possible explanations, it does not logically follow that targeting NR4A1 and NR4A2 by either DIM-3,5 compounds or siRNA should result in reduced protein levels of Sp1 and Sp4, especially since NR4A1 and NR4A2 performed so differently in the GAL4-luciferase assay. There might be a difference in nuclear localization

2) Why is the protein level of beta1-integrin strongly reduced by multiple approaches to targeting NR4A1, NR4A2, Sp1, and Sp4, but its mRNA level is barely changed by DIM-3,5-CI2 treatment in Figure 6D? Although the difference is statistically significant, that is probably because the data come from scRNA-Seq, with thousands of measurements; it is not necessarily biologically significant.

3) There are several questions concerning the evaluation of gene expression in cell lines treated with DIM-3,5-CI2 or siRNA to NR4A1 and/or NR4A2:

  1. Why did the authors use single-cell RNA-Seq for these studies? There is no expected heterogeneity between cells with these treatments, especially with a small-molecule inhibitor (DIM-3,5-CI2), and the analysis does not show consideration of single cells, such as in a Seurat plot. Perhaps there is heterogeneity (such as due to cell-cycle position or the degree of knockdown), and it could have meaningful correlations (such as with reduction in target gene transcription), but the analysis does not mention an attempt to discern this. scRNA-Seq is more expensive than bulk RNA-Seq, and may give less coverage of the transcriptome.
  2. What controls were used? This could have an impact on the next point. For the samples treated with siRNA, it would be essential to use a mock transfection as control, preferably with a scrambled or control siRNA. For DIM-3,5-CI2 there would need to be a control that at least has the same solvent, preferably with an inactive similar molecule (if known).
  3. The most important question is this: Why is there so little overlap in differentially-expressed genes (DEGs) between the conditions?
  4. Especially given how little overlap there is, why was pathway analysis performed using DEGs? Much better would be to use Gene Set Enrichment Analysis.

4) The manuscript should provide the reader with some information about the state of the DIM-3,5 compounds as agents for in vivo use. That this is a possibility is suggested by reference 25. Do they currently appear to have sufficient specificity for clinical use? Or is more optimization required? Are the effects on T-cell exhaustion likely to be direct or indirect?

5) Panel E is not specified in the legend of Figure 2.

Author Response

Comments and Suggestions for Authors – Reviewer 1

1) The authors should provide an explanation for why the protein levels of Sp1 and Sp4 are reduced by DIM-3,5 compounds or siRNA-mediated knockdown of NR4A1 and NR4A2. Although there are multiple possible explanations, it does not logically follow that targeting NR4A1 and NR4A2 by either DIM-3,5 compounds or siRNA should result in reduced protein levels of Sp1 and Sp4, especially since NR4A1 and NR4A2 performed so differently in the GAL4-luciferase assay. There might be a difference in nuclear localization

Response:

  • Previous studies show that CDIM compounds or knockdown of NR4A1 induces ROS in cancer cells and ROS inducers decreased expression of Sp1 and Sp4, and the ROS-Sp degradation pathway has been reviewed (PMID: 29545399). This explanation is now included in the revised manuscript.

2) Why is the protein level of beta1-integrin strongly reduced by multiple approaches to targeting NR4A1, NR4A2, Sp1, and Sp4, but its mRNA level is barely changed by DIM-3,5-CI2 treatment in Figure 6D? Although the difference is statistically significant, that is probably because the data come from scRNA-Seq, with thousands of measurements; it is not necessarily biologically significant.

Response:

  • Knockdown of NR4A1 and NR4A2 usually gives consistent mRNA levels (i.e., after 72 hours and later). However, treatment with CDIMs results in expected decreased mRNA levels however, these change over time: In the present study, β1-integrin mRNA levels were determined at a single time point (i.e. after 24 hr) and this may account for the minimal decreased in mRNA levels. This is now noted in the revised paper.

3) There are several questions concerning the evaluation of gene expression in cell lines treated with DIM-3,5-CI2 or siRNA to NR4A1 and/or NR4A2:

  1. Why did the authors use single-cell RNA-Seq for these studies? There is no expected heterogeneity between cells with these treatments, especially with a small-molecule inhibitor (DIM-3,5-CI2), and the analysis does not show consideration of single cells, such as in a Seurat plot. Perhaps there is heterogeneity (such as due to cell-cycle position or the degree of knockdown), and it could have meaningful correlations (such as with reduction in target gene transcription), but the analysis does not mention an attempt to discern this. scRNA-Seq is more expensive than bulk RNA-Seq, and may give less coverage of the transcriptome.

Response:

  • We thank the reviewer for the thoughtful comment regarding our use of single-cell RNA-seq (scRNA-seq). We agree that for evaluating the effects of DIM-3,5-CIâ‚‚ or NR4A1/NR4A2 knockdown, bulk RNA-seq could have sufficed and may offer deeper transcriptome coverage. However, we selected scRNA-seq in anticipation of using the dataset for broader questions in future studies, such as cell-to-cell transcriptional variability and its modulation under treatment conditions (see: https://pubmed.ncbi.nlm.nih.gov/40113778/).

    While cell lines are generally homogeneous, drug treatments and knockdowns can introduce variability due to factors such as cell cycle stage or differential knockdown efficiency. We now address this more explicitly by including UMAP plots in the revised figure (6C)  to illustrate the transcriptional landscape across individual cells. These plots include cells from the following conditions: Control (scramble), Control (DMSO), NR4A1 KD, NR4A2 KD, NR4A1 & NR4A2 KD, and DIM-3,5-CIâ‚‚ 24 hr. Each knockdown or treatment condition was analyzed relative to its appropriate control.

    Please see the attached PDF to see the revised Fig 6C. UMAP visualization of single cells.

3b. What controls were used? This could have an impact on the next point. For the samples treated with siRNA, it would be essential to use a mock transfection as control, preferably with a scrambled or control siRNA. For DIM-3,5-CI2 there would need to be a control that at least has the same solvent, preferably with an inactive similar molecule (if known).

Response:

  • We thank the reviewer for raising this important point. We apologize for not making the control conditions more explicit in the original submission. As clarified in the newly added UMAP plots, the siRNA knockdown experiments were conducted using a scrambled siRNA as the control, and the DIM-3,5-CIâ‚‚ treatment was compared to a DMSO-treated control. We have now emphasized this information in the revised figure legend and methods section to ensure clarity.

3c. The most important question is this: Why is there so little overlap in differentially-expressed genes (DEGs) between the conditions?

Response:

  • Thank you very much for pointing out this issue—single-cell data can indeed be challenging to analyze and interpret. As each experimental condition has its own appropriate control, cross-comparing differential expression (DE) results between groups must be done carefully.

To explore this further, we compared DE gene sets from two analyses: (1) Control (scramble) vs. NR4A1 & NR4A2 knockdown (comp1), and (2) Control (DMSO) vs. DIM-3,5-CIâ‚‚ 24 hr treatment (comp2). The adjusted p-value–filtered gene lists showed 5,184 DE genes in comp1 and 9,013 in comp2, suggesting different transcriptional responses. Despite the distinct control conditions, we found a substantial overlap between the DE gene sets from the two comparisons, indicating converging biological effects from genetic and chemical perturbation of NR4A1/NR4A2.

Please see the attached PDF to see DE gene list overlap - "First panel gets the whole DE gene list overlap, second panel is the up regulated overlap and third panel is the down regulated overlap."

3d. Especially given how little overlap there is, why was pathway analysis performed using DEGs? Much better would be to use Gene Set Enrichment Analysis.

Response:

  • Both Pathway Analysis of DEG analysis and GSEA are commonly used approaches for interpreting gene expression data in terms of biological processes or pathways. We used DEG-based pathway analysis for a quick overview of major changes. GSEA differs in methodology philosophy and input requirements and may uncover subtler pathway-level trends that DEG lists miss. Our internal test showed that GSEA did not bring any new insights for better interpretation of our data, we therefore we did not include results generated from GSEA analysis. 

4) The manuscript should provide the reader with some information about the state of the DIM-3,5 compounds as agents for in vivo use. That this is a possibility is suggested by reference 25. Do they currently appear to have sufficient specificity for clinical use? Or is more optimization required? Are the effects on T-cell exhaustion likely to be direct or indirect?

Response:

  • We have synthesized several DIM-3,5 analogs and based on in vitro/in vivo studies in several cancer cell lines, we are hoping to generate sufficient data on DIM-3,5-CI2 for FDA approval and a phase I clinical trial. The toxicology/PK/PD needs to be determined, and we currently have a “Stepping Stones” grant from NCI and they are sponsoring generation of some of these data. We think that the effects of DIM-3,5 compounds on T cell exhaustion and expression of cytokines is direct but are still working in this area since indirect effects cannot be excluded.

5) Panel E is not specified in the legend of Figure 2.

Response:

  • This has now been corrected.

Reviewer 2 Report

Comments and Suggestions for Authors

The manuscript presented by Zhang et al. is an excellent scientific work that evaluates the inhibitory effects of the dual ligands NR4A1/A2 (DIM-3.5 and analogs) on the expression of b1-intergrin and G9a genes, regulated by the prooncogenic nuclear receptor 4A (NR4A). This work follows from previous works performed in the same laboratory on a series of bis-indole compounds.

The authors provide a very comprehensive set of methodologies (sections 4.3-4.9) to elucidate the mechanism of action of the dual ligands.

Results provided are very interesting and allow a coherent follow up to elucidate the effects of the dual ligands.

The discussion section is succinct but also meticulously organized. The same for the Materials and Methods section; very well organized with clear descriptions of all materials and methodologies.

Author Response

Comments and Suggestions for Authors – Reviewer 2

The manuscript presented by Zhang et al. is an excellent scientific work that evaluates the inhibitory effects of the dual ligands NR4A1/A2 (DIM-3.5 and analogs) on the expression of b1-intergrin and G9a genes, regulated by the prooncogenic nuclear receptor 4A (NR4A). This work follows from previous works performed in the same laboratory on a series of bis-indole compounds.

The authors provide a very comprehensive set of methodologies (sections 4.3-4.9) to elucidate the mechanism of action of the dual ligands.

Results provided are very interesting and allow a coherent follow up to elucidate the effects of the dual ligands.

The discussion section is succinct but also meticulously organized. The same for the Materials and Methods section; very well organized with clear descriptions of all materials and methodologies.

Response:

No negative comments to address. Thank you for reviewing.